# A large gene family in fission yeast encodes spore killers that subvert Mendel's law

Wen Hu[1], Zhao-Di Jiang[1,2], Fang Suo[1], Jin-Xin Zheng[1], Wan-Zhong He[1], Li-Lin Du[1]*

[1]National Institute of Biological Sciences, Beijing, China; [2]PTN Graduate Program, School of Life Sciences, Tsinghua University, Beijing, China

**Abstract** Spore killers in fungi are selfish genetic elements that distort Mendelian segregation in their favor. It remains unclear how many species harbor them and how diverse their mechanisms are. Here, we discover two spore killers from a natural isolate of the fission yeast *Schizosaccharomyces pombe*. Both killers belong to the previously uncharacterized *wtf* gene family with 25 members in the reference genome. These two killers act in strain-background-independent and genome-location-independent manners to perturb the maturation of spores not inheriting them. Spores carrying one killer are protected from its killing effect but not that of the other killer. The killing and protecting activities can be uncoupled by mutation. The numbers and sequences of *wtf* genes vary considerably between *S. pombe* isolates, indicating rapid divergence. We propose that *wtf* genes contribute to the extensive intraspecific reproductive isolation in *S. pombe*, and represent ideal models for understanding how segregation-distorting elements act and evolve.

## Introduction

Mendel's law of equal segregation stipulates that paternal and maternal alleles of a gene should have an equal chance of being transmitted to progenies. This law guarantees a fair competition between different alleles and enables beneficial ones to prevail during natural selection. Meiotic drivers, a type of selfish genetic element, break Mendel's law by skewing transmission ratios to their advantage, and thus can spread in a population even when having a deleterious effect on organismal fitness (*Lindholm et al., 2016*; *Werren, 2011*). The term 'meiotic drive' was initially coined to describe segregation distortion resulting from preferential inclusion in the gamate during asymmetric female meiosis but has now been used more broadly to include biased transmission caused by postmeiotic mechanisms. In fact, some of the best-known meiotic drivers, such as *Segregation Distorter* in *Drosophila* and the *t* haplotype in mouse (*Lyon, 2003*; *Larracuente and Presgraves, 2012*), act postmeiotically to disable male gametes (sperms) that do not inherit them. This type of meiotic driver, called gamete killer, exists in animals, plants, and fungi (*Burt and Trivers, 2006*). Fungal gamete killers, or spore killers, have been found in several filamentous ascomycetes, most notably *Neurospora* and *Podospora anserina* (*Dalstra et al., 2003*; *Grognet et al., 2014*; *Hammond et al., 2012*; *Turner and Perkins, 1979*). It is unclear how widespread spore killers are among fungal species.

The fission yeast *Schizosaccharomyces pombe* is a prominent model organism for molecular and cell biology and has been increasingly used to study natural variation and genome evolution (*Brown et al., 2011*; *Hu et al., 2015*; *Jeffares et al., 2015*; *Rhind et al., 2011*). *S. pombe* natural isolates, which are nearly all haploids and have pair-wise nucleotide differences of less than 1% (*Jeffares et al., 2015*; *Rhind et al., 2011*), can readily mate with each other to form hybrid diploids, but the viability of spores derived from inter-isolate crosses is often below 5% and in many instances

*For correspondence: dulilin@nibs.ac.cn

**eLife digest** During evolution, new species emerge when individuals from different populations of similar organisms no longer breed with each other, or when the offspring produced if they do breed are sterile. This process is known as "reproductive isolation" and, for over 100 years, evolutionary biologists have tried to better understand how this process happens.

Animals, plants and fungi produce sex cells – known as gametes – when they are preparing to reproduce. These cells are made when cells containing two copies of every gene in the organism divide to produce new cells that each only have one copy of each gene. Therefore, a particular gene copy usually has a 50% chance of being carried by an individual gamete. There are genes that selfishly increase their chances of being transmitted to the next generation by destroying the gametes that do not carry them. These "gamete killer" genes reduce the fertility of the organism and lead to reproductive isolation.

Fission yeast is a fungus that is widely used in research. There are different strains of fission yeast that are reproductively isolated from each other, but it is not known whether gamete killers are responsible for this isolation. To address this question, Hu et al. investigated the causes of reproductive isolation in fission yeast.

The experiments identified two gamete killers, referred to as *cw9* and *cw27*. Both genes belong to the *wtf* gene family. Each gene is believed to encode two different proteins, one that acts as a poison and one that acts as an antidote. The poison is capable of killing all gametes, but the antidote protects the cells that contain the gamete killer gene. Further experiments show that the antidote produced by one of the gamete killer genes cannot protect cells against the poison produced by the other gene.

A separate study by Nuckolls et al. found that another member of the *wtf* family also acts as a gamete killer in fission yeast. Together, these findings shed new light on the causes of reproductive isolation, and will contribute to deeper understanding of speciation and evolution in general.

under 1% (*Gutz and Doe, 1975*; *Kondrat''eva and Naumov, 2001*; *Naumov et al., 2015*). This is in stark contrast to *Saccharomyces cerevisiae*, whose natural isolates have a similar level of nucleotide diversity but much better spore viability when inter-crossed (*Hou et al., 2014*). One explanation for the within-species hybrid sterility in *S. pombe* is chromosomal rearrangement (*Avelar et al., 2013*; *Brown et al., 2011*; *Zanders et al., 2014*). However, because one rearrangement reduces spore viability at most by half (*Avelar et al., 2013*; *Hou et al., 2014*), other factors are likely in play. It was shown recently that when the *S. pombe* laboratory strain, which was isolated from French grape juice (*Hu et al., 2015*; *Osterwalder, 1924*), was crossed to a strain isolated from fermented tea (initially called *Schizosaccharomyces kambucha* [*Singh and Klar, 2003*], later renamed *S. pombe var. kambucha* [*Rhind et al., 2011*]), at least three spore killers contributed to hybrid sterility (*Bomblies, 2014*; *Zanders et al., 2014*), but the identities of the killer genes were unknown.

In this study, through investigating the causes of intraspecific hybrid sterility of fission yeast, we uncovered the molecular identities of two active spore killers, which both belong to the *wtf* gene family. We analyzed their killing behaviors in both native and non-native genomic contexts. Furthermore, we performed comparative genomic analysis of *wtf* genes and revealed interesting patterns of divergence among them.

## Results

We have previously studied CBS5557, an *S. pombe* strain isolated from grapes in Spain (*Hu et al., 2015*; *Rankine and Fornachon, 1964*). Like most *S. pombe* isolates, CBS5557 differs from the laboratory strain, whose genome is the *S. pombe* reference genome, in a 2.23 Mb pericentric inversion on chromosome I (*Brown et al., 2011*). After introducing the same inversion into the laboratory strain (*Hu et al., 2015*), the viability of spores derived from the cross between the laboratory strain and CBS5557 increased from 14% to 22% (*Figure 1—figure supplement 1* and *Supplementary file 1*). To uncover the causes of the remaining sterility, we performed next-generation sequencing (NGS)-assisted bulk segregant analysis (*Hu et al., 2015*) and found that among the viable progenies

from the cross, SNP alleles of the laboratory strain (reference alleles) on chromosome III were markedly under-represented (*Figure 1A*). Two broad transmission distortion peaks, one on the left arm (hereafter referred to as the left peak) and one on the right arm (hereafter referred to as the right peak), were observed, suggesting that there are at least two spore killer genes on chromosome III of CBS5557.

To narrow down the genomic regions containing the spore killer genes, we performed multiple rounds of backcross and obtained strains with only a portion of chromosome III originating from CBS5557 and the rest of the genome coming from the laboratory strain (see 'Materials and methods'). We selected two backcrossed strains that retained the ability to skew inheritance of markers inserted in the left and right peak regions, respectively. These two strains, referred to as backcrossed-1 and backcrossed-2, respectively, were crossed to the laboratory strain and viable progenies were subjected to NGS-assisted bulk segregant analysis, which revealed the locations of the CBS5557 genomic segments present in the two backcrossed strains and confirmed the allele transmission bias (left-side plots in *Figure 1B and C*).

Through inspecting the Illumina and PacBio sequencing data of the CBS5557 genome, we discovered that, within the CBS5557 genomic segments in the two backcrossed strains, sequences most divergent from the reference genome are those containing *wtf* family genes (*Figure 1D and E*). *wtf* (for with Tf LTRs) genes were so named because they are often in close proximity to solo long terminal repeats (LTRs) derived from Tf retrotransposons (*Bowen et al., 2003*; *Wood et al., 2002*). *wtf* genes are predicted to contain multiple introns and encode multi-transmembrane proteins and have been shown to be highly up-regulated transcriptionally during meiosis and sporulation by large-scale analyses (*Bowen et al., 2003*). No homologs of *wtf* genes have been found in any other species, including other *Schizosaccharomyces* species (*Rhind et al., 2011*). We deleted from the backcrossed-1 strain a *wtf* gene locating at the position of *wtf9* in the reference genome but exhibiting a low sequence identity (42%) to *wtf9* in the 5' portion (*Figure 1D* and *Figure 1—figure supplement 2*) and found that the deletion notably reduced the segregation distortion (right-side plot in *Figure 1B*), suggesting that this *wtf* gene, which we named *cw9* (the nomenclature of *wtf* genes in the CBS5557 genome will be explained later in this paper), is a spore killer. The residual allele frequency bias is probably due to additional spore killer gene(s) in the CBS5557 genomic segments present in the backcrossed-1 strain. Within the CBS5557 genomic segment in the backcrossed-2 strain, we found a *wtf* gene absent in the reference genome, possibly owing to a deletion event mediated by homologous recombination between LTRs (*Figure 1E* and *Figure 1—figure supplement 3*). Removing this *wtf* gene, which we named *cw27*, from the backcrossed-2 strain completely abolished segregation distortion (right-side plot in *Figure 1C*).

To determine how *cw9* and *cw27* behave in their native strain background, we generated $h^+/h^-$ CBS5557 diploid strains containing homozygous or heterozygous deletion of *cw9* or *cw27*, and performed tetrad analysis to evaluate spore viability (*Figure 2A* and *Supplementary file 1*). Homozygous deletion of *cw9* or *cw27* had no effect on spore viability (*Figure 2A* and *Figure 2—figure supplement 1*). In contrast, heterozygous deletion of either *cw9* or *cw27* caused a significant reduction of spore viability, which resulted mainly from the death of spores carrying the deletion (*Figure 2B*). When *cw9* and *cw27* were both in the heterozygous state, a more severe loss of spore viability was observed, with the double deletion mutant spores suffering a particularly high level of death (*Figure 2A and B*, and *Supplementary file 1*). These data indicate that *cw9* and *cw27* can act as spore killers in the self-cross of CBS5557 when either of them is in the heterozygous state, and inheriting one killer can prevent its killing effect but not that of the other killer.

To determine whether *cw9* and *cw27* can cause spore killing in a non-native genomic context, we constructed integrating plasmids carrying *cw9* or *cw27*, integrated the plasmids at various genomic locations in the laboratory strain, and subjected the resulting strains to crosses. In crosses where only one of the parental strains harbored *cw9* or *cw27*, spores inheriting the killer had normal viability, whereas spores not inheriting the killer suffered significant loss of viability, regardless of the genomic location of the killer (*Figure 3A* and *Figure 3—figure supplement 1*). We then performed crosses in which both parental strains harbored a killer at the *leu1* locus (*Figure 3B* and *Figure 3—figure supplement 2*). When both parental strains contained the same killer, spore viability was normal. In contrast, when one parental strain had *cw9* and the other parental strain had *cw27*, spores inheriting either *cw9* or *cw27* experienced a dramatic loss of viability. These results demonstrate that

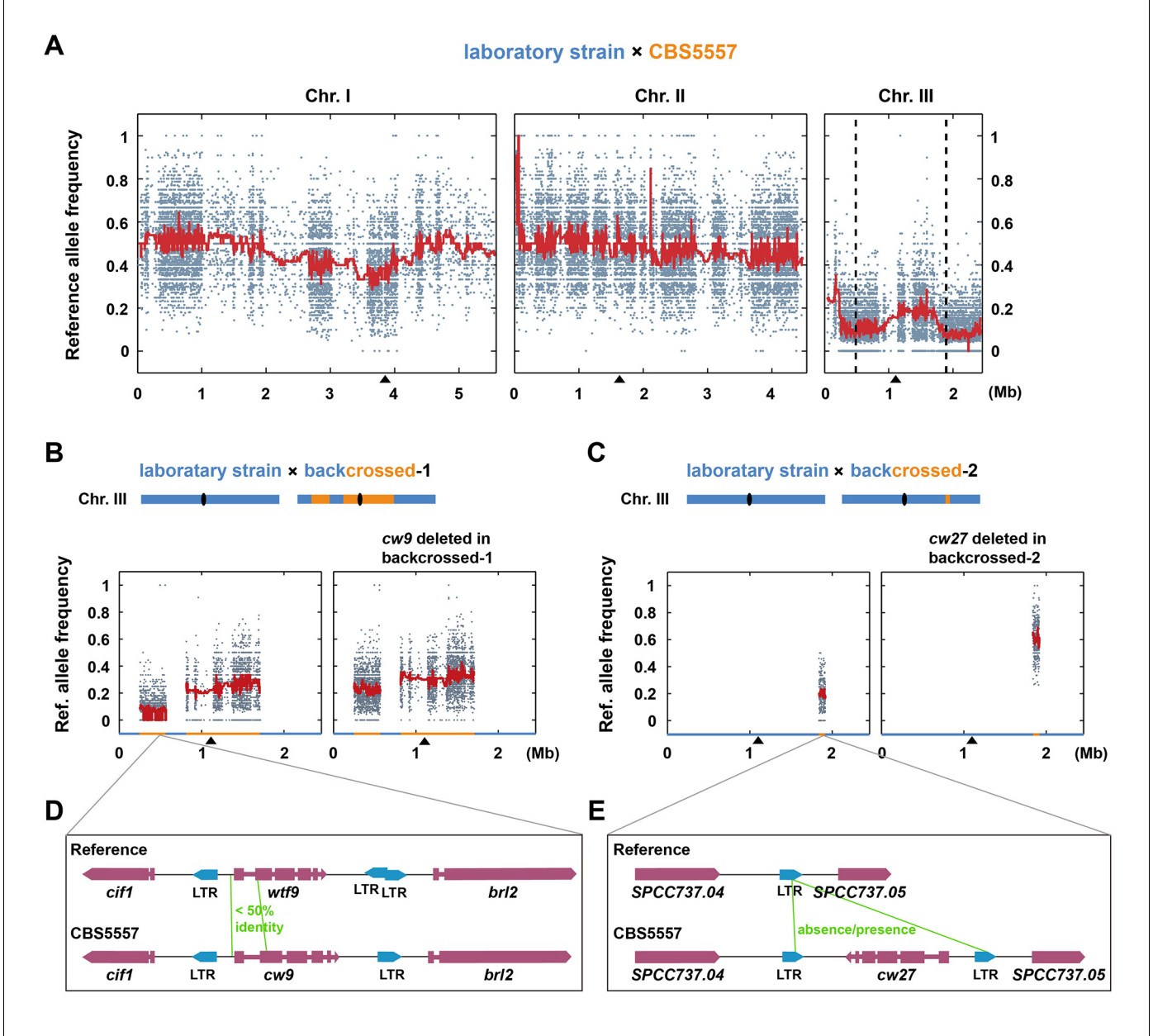

**Figure 1.** The identification of *cw9* and *cw27* genes as spore killers. (**A**) NGS-assisted bulk segregant analysis of viable progenies from a cross between DY9974, a heterothallic derivative of CBS5557, and DY8531, a laboratory strain harboring the 2.23 Mb chromosome I inversion (*Hu et al., 2015*). Reference allele frequencies at SNP positions are plotted as dots. Red trend lines are based on a rolling median calculation. Two vertical dashed lines denote the positions of *cw9* and *cw27* genes on chromosome III. Black triangles mark the positions of centromeres. (**B and C**) Bulk segregant analysis of viable progenies from crosses between a laboratory strain and the backcrossed-1 strain, and between a laboratory strain and the backcrossed-2 strain, respectively. Blue and orange colored segments in the diagrams and on the X-axes of the plots denote chromosome III regions with reference genome sequence and those with CBS5557 genome sequence, respectively, in the two backcrossed strains. (**D and E**) Schematics of the regions surrounding *cw9* and *cw27* in the CBS5557 genome and the corresponding regions in the reference genome. Gene structures of *cw9* and *cw27* were predicted using the AUGUSTUS web server (*Stanke et al., 2008*). Solo LTRs were annotated based on BLAST analysis.

The following figure supplements are available for figure 1:

**Figure supplement 1.** Hybrid sterility between CBS5557 and the laboratory strain can be partially rescued by eliminating the difference in the 2.23 Mb chromosome I inversion.

**Figure supplement 2.** Alignment of the 5' portions of the *cw9* gene of CBS5557 and the *wtf9* gene of the reference genome.

*Figure 1 continued on next page*

*Figure 1 continued*

**Figure supplement 3.** Alignment of the two directly oriented LTRs flanking *cw27* in the CBS5557 genome (top and bottom sequences, respectively) and the LTR at the corresponding location in the reference genome (middle sequence).

*cw9* and *cw27* can act independently of genomic context and confirm that they do not exhibit mutual resistance.

To observe the ultrastructural details of the spore killing process, we performed electron microscopy analysis on laboratory-background $h^+/h^-$ diploid cells undergoing synchronous meiosis and sporulation induced by a shift to a nitrogen-free synthetic sporulation medium (SSL-N medium) (*Figure 3C*). Two diploid strains were used: a strain heterozygous for a *cw27*-containing plasmid integrated at the *his3* locus and a control strain heterozygous for an empty vector integrated at the same locus. 12 hr after the shift, *cw27*-containing cells formed four-spored asci indistinguishable from those formed by the control cells. At this time point, organelles inside spores were clearly visible. 18 hr after the shift, the cytoplasm of spores derived from the control cells became electron-dense and organelles were no longer discernable, owing to a little-understood spore maturation process (*Yoo et al., 1973*). Interestingly, at the same time point, asci derived from the *cw27*-containing cells often exhibited a 2:2 pattern of spore morphology: two spores in an ascus looked like the matured spores in the control sample, whereas the other two spores exhibited cytoplasm with a lower electron density and visible organelles, an electron-lucent spore wall, and frequently a non-spherical shape. These data suggest that killer-affected spores fail to undergo a proper maturation process.

*cw9* and *cw27* are both flanked by solo LTRs, and upstream of their predicted start codons they share a nearly identical 288-bp-long sequence (hereafter referred to as the conserved_up sequence) (*Figure 4—figure supplement 1*), which is also conserved upstream of the coding sequences of many other *wtf* genes (*Bowen et al., 2003*). To determine whether these and other sequence features are important for spore killing, we performed truncation analysis (*Figure 4A* and *Figure 4—figure supplement 2*). The truncated versions were named Ta, Tb, Tc, Td, and Te. Their effects on spore viability in a heterozygous cross of laboratory strains were examined. The Ta versions, which lack the LTRs, caused spore killing phenotypes similar to those caused by longer constructs containing the flanking LTRs, indicating that the LTRs are not important. Interestingly, the Tb versions missing the sequence between the upstream LTR and the conserved_up sequence caused spores containing the killer to suffer a moderate viability loss, suggesting that these truncated killer genes partially lost the ability to protect killer-containing spores from being harmed. Removing from the Ta versions, the sequence downstream of the predicted stop codon (Tc versions) did not have a notable effect. Removing from the Tc versions, the last 10 amino-acid-coding codons (Td versions) greatly diminished spore killing, whereas removing from the Tc versions the sequences upstream of the start codons (Te versions) resulted in more indiscriminate killing than that caused by the Tb versions. These results suggest that intact C termini are important for the killing activity, and the upstream sequences are important for the protecting activity.

To determine whether the Td versions, which are largely devoid of the killing activity, can still protect, we performed crosses in which a strain with a Td-version-containing plasmid integrated at the *ars1* locus was crossed to a strain with a Tc-version-containing plasmid integrated at the *leu1* locus, and found that the Td versions of both *cw9* and *cw27* can effectively protect spores harboring them from being killed by the respective Tc-version killers (*Figure 4B* and *Figure 4—figure supplement 3*). Thus, perturbing the C termini resulted in separation-of-function mutants that retain the protecting activity but can no longer kill. Such kind of mutants, if present in natural populations, would behave similarly to the resistant alleles of other previously studied gamete killers, and may also represent an intermediate state toward the extinction of a killer (*Burt and Trivers, 2006*).

In the reference genome, there are 25 *wtf* genes at 20 locations (15 singletons and five tandem pairs) (*Bowen et al., 2003*) (*Figure 5A and B*). Using PacBio sequencing, we found that in the CBS5557 genome, these 20 locations are occupied by 29 *wtf* genes (12 singletons, seven pairs, and one triplet), with four locations containing an extra *wtf* gene compared to the reference genome

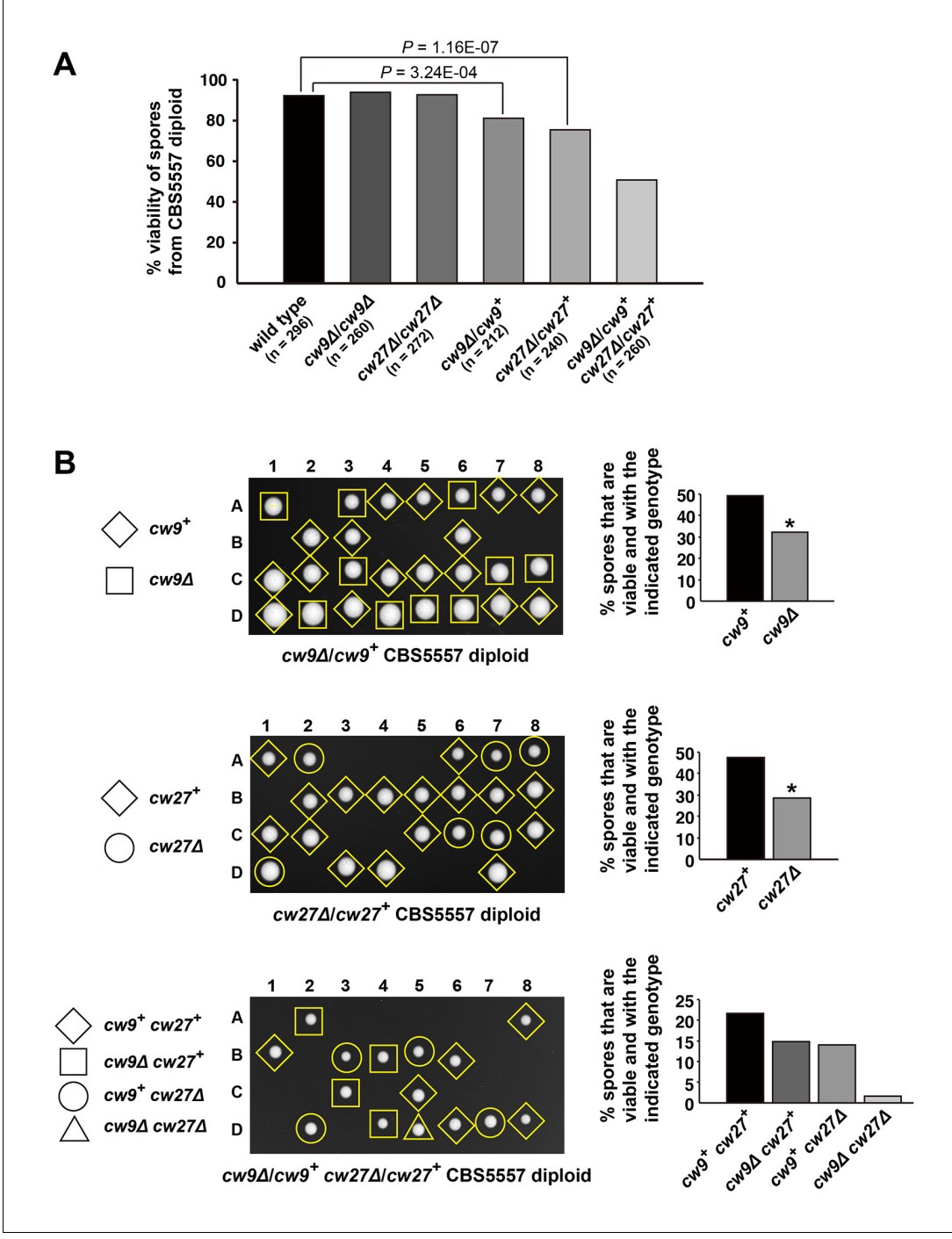

**Figure 2.** *cw9* and *cw27* act as spore killers when in the heterozygous state in the CBS5557 background. (**A**) Heterozygous deletion but not homozygous deletion of *cw9* or *cw27* in CBS5557 $h^+$/$h^-$ diploid caused spore viability loss. Spore viability was measured using tetrad analysis. Representative tetrads are shown in *Figure 2— figure supplement 1* and in panel B. p-values were calculated using Fisher's exact test. Numerical data are provided in *Supplementary file 1*. (**B**) Among the spores derived from CBS5557 diploids with heterozygous deletion, loss of viability mainly occurred to spores with deletion. Asterisks indicate significant deviation from 50% (p=1.94E-7 and 1.42E-11 for *cw9Δ* spores from *cw9Δ/cw9⁺* diploid and *cw27Δ* spores from *cw27Δ/cw27⁺* diploid, respectively, exact binomial test). Numerical data are provided in *Supplementary file 1*.

The following figure supplement is available for figure 2:

*Figure 2 continued on next page*

*Figure 2 continued*

**Figure supplement 1.** Representative tetrads from a wild type CBS5557 $h^+/h^-$ diploid strain (DY21782), a CBS5557 $h^+/h^-$ diploid strain with homozygous *cw9* deletion (DY21838), and a CBS5557 $h^+/h^-$ diploid strain with homozygous *cw27* deletion (DY21842).

(*Figure 5A and B*). We systematically named these 29 genes according to their syntenic relationship with the 25 *wtf* genes in the reference genome, with each name consisting of the prefix *cw*, a number (from 1 to 25), and a suffix for genes at locations where extra genes exist in the CBS5557 genome (*Figure 5A and B*). In addition, there are three CBS5557 singleton *wtf* genes without syntenic counterparts in the reference genome and we named them *cw26*, *cw27*, and *cw28* according to their order in the genome (*Figure 5A*). Like *cw27*, *cw26* and *cw28* also appeared to have been lost in the laboratory strain through LTR-mediated recombination (*Figure 5—figure supplement 1*), suggesting that this is a common mode of *wtf* gene turnover.

Among the 12 CBS5557 singleton *wtf* genes with syntenic counterparts in the reference genome, *cw9* and five others exhibit exceptionally high DNA sequence diversity (<85% identity) from their counterparts in the reference genome (*Figure 1D*, *Figure 1—figure supplement 2*, and *Figure 5—figure supplement 2*). Given that the genome-wide nucleotide difference between CBS5557 and the laboratory strain is only 3.1 SNPs/kb (*Hu et al., 2015*), *wtf* genes must have diverged at a much faster pace than average genes.

To examine the phylogenetic relationship among all 57 *wtf* genes of the two genomes, we performed a maximum likelihood analysis (*Figure 5C*). In the resulting phylogenetic tree, we used colored rectangles to highlight 16 pairs of genes, with each pair consisting of phylogenetic neighbors, defined as two genes separated by a single internal node that has an ultrafast bootstrap support value >= 95% (*Figure 5C*). Interestingly, only 11 of these 16 pairs conform to typical orthologous relationship where two members of a pair are syntenic (highlighted by green rectangles in *Figure 5C*), whereas the five other pairs each consist of two genes that are non-syntenic (highlighted by yellow rectangles in *Figure 5C*). This pattern suggests that ectopic gene conversion (also called non-allelic or interlocus gene conversion) may contribute to the inter-isolate difference of *wtf* genes (*Chen et al., 2007*; *Petes and Hill, 1988*). Interestingly, genes belonging to the same gene cluster (pair or triplet), despite being subject more strongly to the homogenizing effect of gene conversion due to their physical proximity, are without exception very distinct from each other. One possible explanation is that diversifying selection may have prevented sequence homogenization within gene clusters.

The substantial difference in the numbers and sequences of *wtf* gene between the laboratory strain and CBS5557 suggests that rapid sequence divergence and independent loss/gain of *wtf* genes have resulted in different *S. pombe* natural isolates harboring distinct sets of active spore killers, which become reproductive barriers during inter-isolate crosses and contribute to the extensive and severe hybrid sterility within this species.

## Discussion

Only a small number of gamete killers have been characterized at the molecular level. Two well-studied gamete killers in animals, *Segregation Distorter* in *Drosophila* and the *t* haplotype in mouse, both contain at least two key loci, a killer locus and a target locus (*Lyon, 2003*; *Burt and Trivers, 2006*; *Larracuente and Presgraves, 2012*; *Lindholm et al., 2016*; *Bauer et al., 2012*). In fungi, spore killers of *Neurospora* also appear to comprise of two genes, *rsk* (*resistant to Spore killer*) and a yet-to-be-identified killer gene, which are proposed to act together in a fashion similar to the toxin-antitoxin (TA) systems of bacterial plasmids, in which a stable toxin and a labile antitoxin together ensure that bacterial cells cannot lose the plasmid expressing them (*Hammond et al., 2012*; *McLaughlin and Malik, 2017*; *Saupe, 2012*). The involvement of more than one gene renders these gamete killer systems vulnerable to recombination that may separate the component genes, and as a consequence, they have all evolved recombination-suppressing features such as chromosomal inversions (*Burt and Trivers, 2006*; *Harvey et al., 2014*). On the other hand, two types of single-gene spore killers, [Het-s] prion and the *Spok* genes, have been found in the filamentous fungus

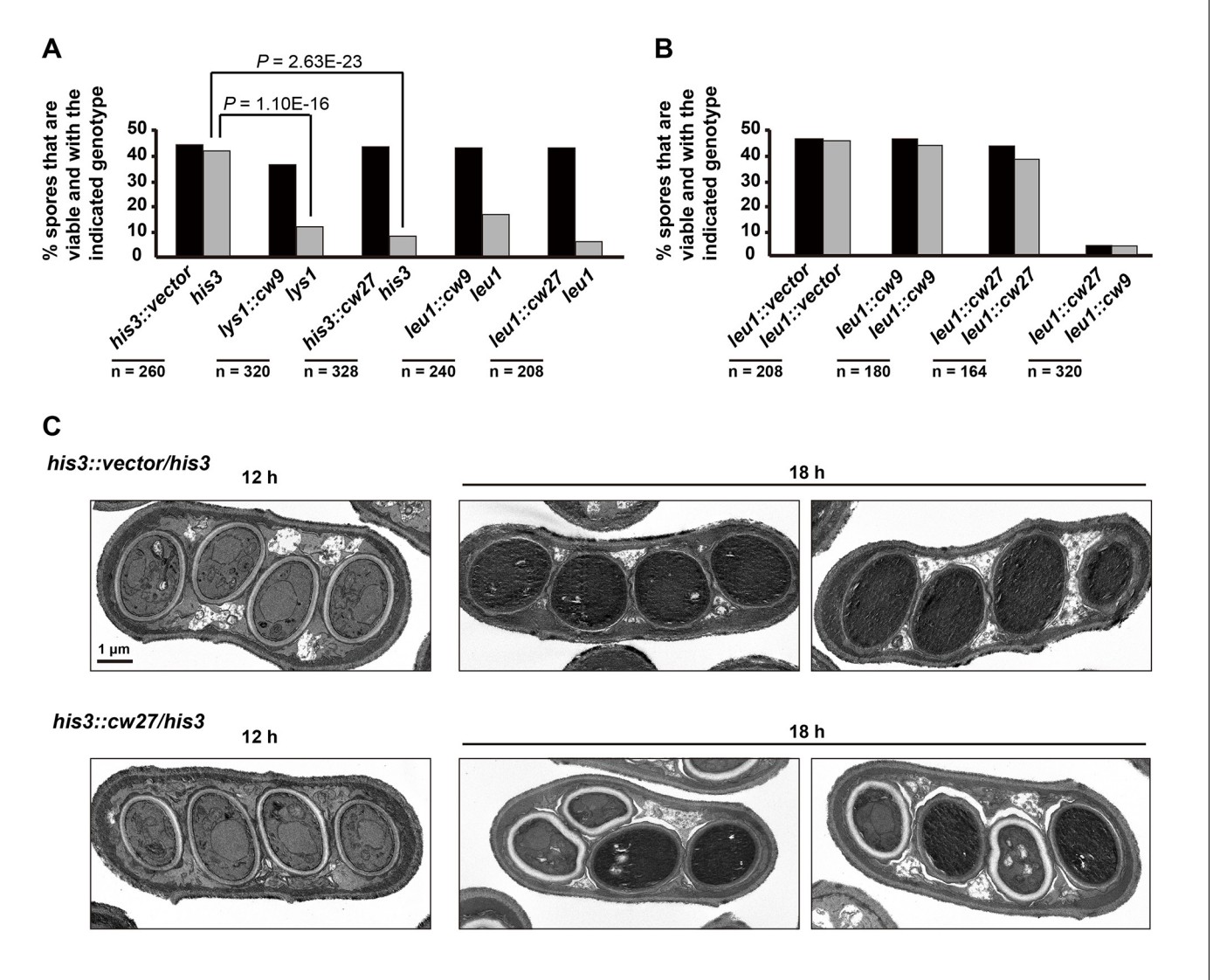

**Figure 3.** *cw9* and *cw27* act as spore killers when inserted into the genome of the laboratory strain. (**A**) In crosses of laboratory strains, when only one parental strain had an insertion of *cw9* or *cw27* at *his3*, *lys1*, or *leu1* locus, spores without the insertion suffered viability loss. Representative tetrads are shown in *Figure 3—figure supplement 1*. p-values were calculated using Fisher's exact test. Numerical data are provided in *Supplementary file 1*. (**B**) In crosses of laboratory strains, when both parental strains had an insertion of *cw9* or *cw27* at the *leu1* locus, spore viability was normal when parents had the same killer, but was severely low when parents had different killers. The two parental alleles were distinguished by *leu1*-linked antibiotic resistance markers. Representative tetrads are shown in *Figure 3—figure supplement 2*. Numerical data are provided in *Supplementary file 1*. (**C**) Electron microscopy analysis of laboratory-background *h⁺/h⁻* diploid cells undergoing synchronous meiosis and sporulation.

The following figure supplements are available for figure 3:

**Figure supplement 1.** Representative tetrads from laboratory-background crosses in which only one of the parental haploid strains had a vector or a killer-containing plasmid integrated at the *his3*, *lys1*, or *leu1* locus.

**Figure supplement 2.** Representative tetrads from three laboratory-background *h⁺/h⁻* diploid strains homozygous for plasmid integration at the *leu1* locus and a laboratory-background *h⁺/h⁻* diploid strain heterozygous for plasmid integration at the *leu1* locus, with one allele containing *cw9* and the other allele containing *cw27*.

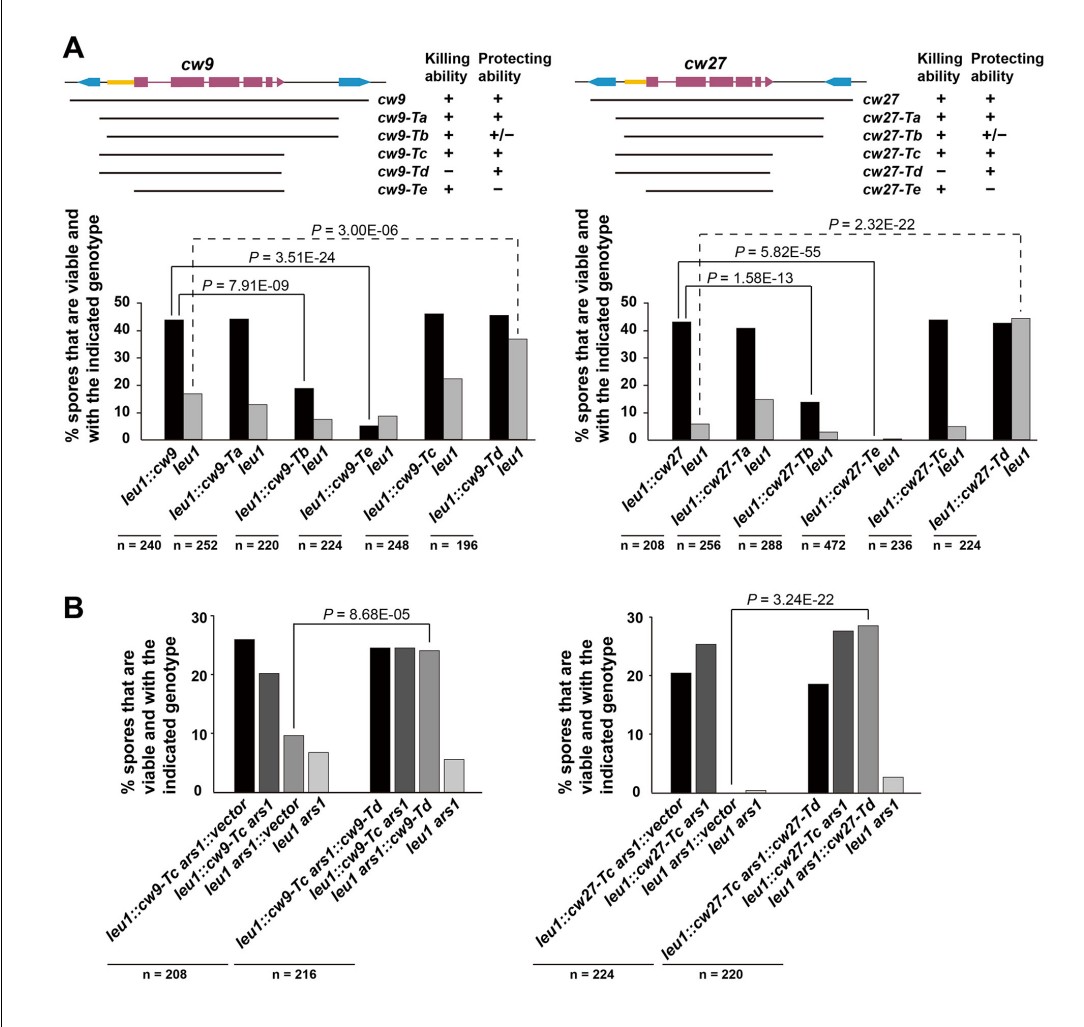

**Figure 4.** Sequence requirement for the killing and the protecting activities of *cw9* and *cw27*. (**A**) Truncation analysis to assess the involvement of the 5' and 3' sequences of *cw9* and *cw27* in spore killing. In the diagrams on top, blue arrows represent LTRs and yellow bars represent the conserved_up sequence. Representative tetrads are shown in **Figure 4—figure supplement 1**. p-values were calculated using Fisher's exact test. Numerical data are provided in **Supplementary file 1**. (**B**) The Td versions of *cw9* and *cw27* are able to effectively protect against killing despite their lack of killing activity. Representative tetrads are shown in **Figure 4—figure supplement 2**. p-values were calculated using Fisher's exact test. Numerical data are provided in **Supplementary file 1**.

The following figure supplements are available for figure 4:

**Figure supplement 1.** Alignment of the 5' portions of *cw9* and *cw27*.

**Figure supplement 2.** Representative tetrads from laboratory-background $h^+/h^-$ diploid strains heterozygous for plasmid integration at the *leu1* locus.

**Figure supplement 3.** Representative tetrads from laboratory-background $h^+/h^-$ diploid strains heterozygous for a Tc-version killer-containing plasmid integrated at the *leu1* locus on chromosome II, and also heterozygous for a vector or a Td-version killer-containing plasmid integrated at the *ars1* replication origin region upstream of the *hus5* gene on chromosome I.

*Podospora anserina* (**Dalstra et al., 2003**; **Grognet et al., 2014**). The killing activity of the [Het-s] prion requires the presence of the HET-S protein, whereas the *Spok* genes can act autonomously. Fission yeast *wtf* killers share two prominent features with the *Spok* killers: being composed of a single predicted coding sequence and acting in a self-sufficient manner. These features arguably make *wtf* and *spok* killers the most compact and efficient forms of gamete killers.

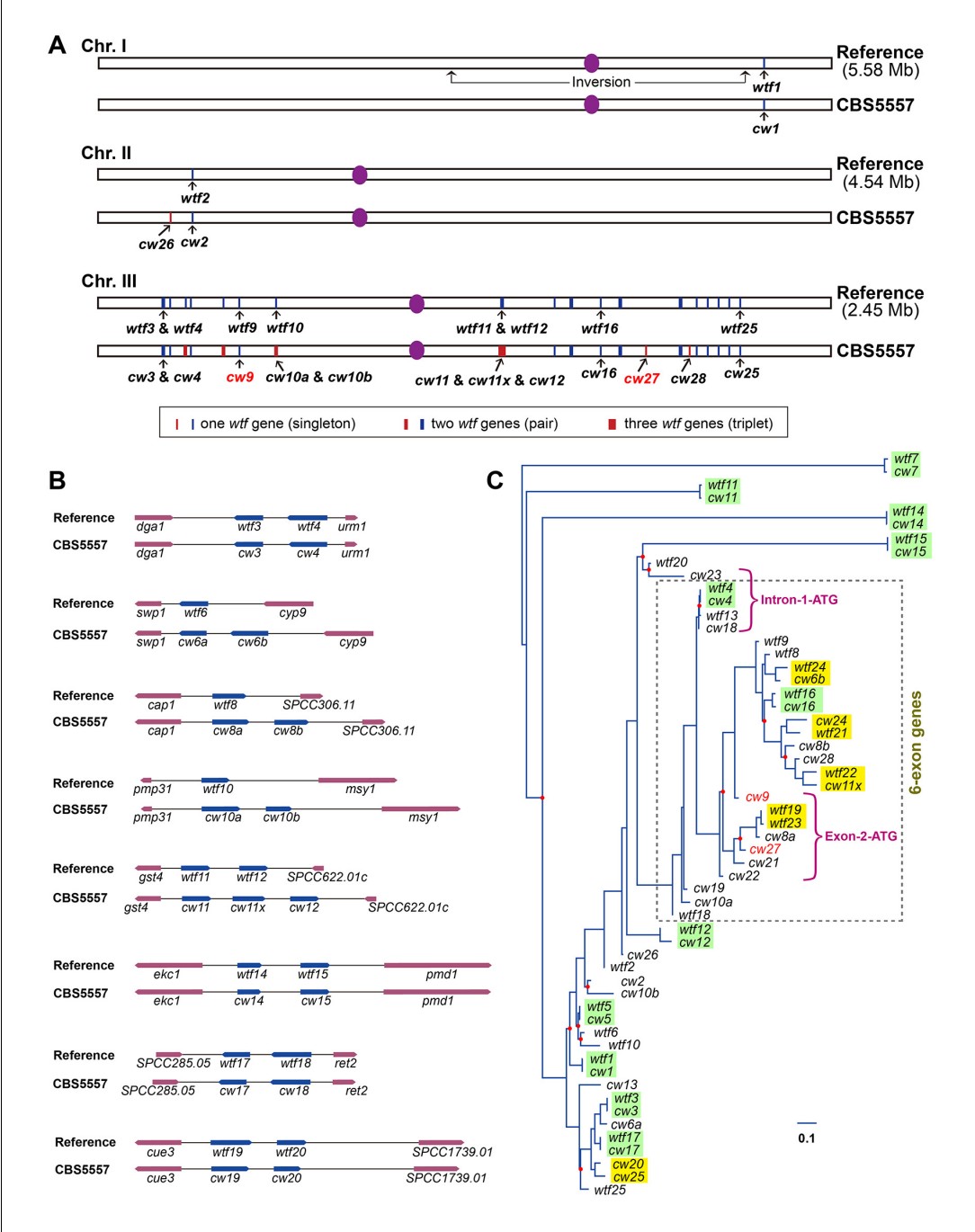

**Figure 5.** *wtf* genes vary both in numbers and sequences between the reference genome and the CBS5557 genome. (**A**) Genomic locations of *wtf* genes in the reference genome and in the CBS5557 genome. The 25 *wtf* genes in the reference genome have been named according to their order in the genome (***Bowen et al., 2003***). Their locations are depicted as 20 blue vertical bars, including five thick bars denoting five tandem pairs. Our PacBio sequencing analysis revealed that in the CBS5557 genome, there are 32 *wtf* genes, whose locations are depicted as 23 vertical bars, including three thin red bars at locations where no *wtf* genes exist in the reference genome and four thick red bars at locations where compared to the reference genome one extra *wtf* gene is found. The *wtf* genes in the CBS5557 genome are named with the prefix *cw* and a number from the name of the syntenic gene in the reference genome. Two genes of a tandem pair corresponding to a singleton in the reference genome are distinguished using the suffixes a and b. Among the three genes of the triplet, two are named *cw11* and *cw12* based on their homology to *wtf11* and *wtf12*, respectively, and the gene situated between *cw11* and *cw12* is named *cw11x*. Genes at new locations are named *cw26*, *cw27*, and *cw28*. Chromosome lengths are not drawn to scale. (**B**) Diagrams depicting the eight genomic locations with more than one *wtf* gene in at least one of the two genomes. Genes are shown as arrows. Introns are not shown. *wtf17* is depicted not according to its annotation at PomBase, with its 5' boundary revised based on sequence alignment. (**C**) Maximum likelihood phylogenetic tree of 57 *wtf* genes of the reference genome and the CBS5557 genome. DNA sequences including the conserved_up regions,

*Figure 5 continued on next page*

*Figure 5 continued*

predicted coding sequences, and associated introns were aligned using the L-INS-i iterative refinement algorithm of MAFFT (*Katoh and Standley, 2014*) (*Figure 5—source data 1*). Maximum likelihood analysis was performed using IQ-TREE (*Nguyen et al., 2015*). The tree was rooted by midpoint rooting (*Hess and De Moraes Russo, 2007*). Red dots on nodes indicate IQ-tree-calculated ultrafast bootstrap (UFBoot) support values < 95%. Colored rectangles highlight phylogenetic neighbors. Two genes are considered phylogenetic neighbors if they are separated by a single internal node with a support value >= 95%. Green rectangles indicate the 11 pairs of neighbors each composed of a reference *wtf* gene and a syntenic CBS5557 *wtf* gene. Yellow rectangles indicate the five pairs each composed of two *wtf* genes locating at different genomic positions. Magenta brackets denote the 12 genes that share a 150 bp sequence within the predicted intron 1 (see *Figure 5—figure supplement 3*). These 12 genes are divided into two subtypes, Intron-1-ATG genes and Exon-2-ATG genes, based on the locations of the first ATG codons downstream of the 150-bp-long sequence. The brown dashed box indicates genes with six exons. Scale bar, 0.1 nucleotide substitutions per nucleotide site.

The following source data and figure supplements are available for figure 5:

**Source data 1.** MAFFT-aligned DNA sequences of 57 *wtf* genes of the reference genome and the CBS5557 genome.

**Source data 2.** Gene structure predictions for 57 *wtf* genes of the reference genome and the CBS5557 genome.

**Source data 3.** MAFFT-aligned amino acid sequences of the predicted protein products of 57 *wtf* genes of the reference genome and the CBS5557 genome.

**Figure supplement 1.** Schematics of how the three CBS5557-only singleton *wtf* genes may have been lost in the laboratory strain through LTR-mediated recombination.

**Figure supplement 2.** Schematics depicting the high levels of diversity between five singleton CBS5557 *wtf* genes and their counterparts in the reference genome.

**Figure supplement 3.** *cw9*, *cw27*, and 10 other *wtf* genes in the reference genome and the CBS5557 genome share a 150 bp conserved sequence in the predicted intron 1.

**Figure supplement 4.** The predicted protein products of 6-exon-containing *wtf* genes in the reference genome and the CBS5557 genome.

**Figure supplement 5.** The predicted protein products of 5-exon-containing *wtf* genes in the reference genome and the CBS5557 genome.

Despite the similarities they share, *wtf* and *spok* killers are very different entities. Firstly, *wtf* genes only exist in a single fission yeast species *S. pombe*, indicating that they probably have arisen after the divergence of *S. pombe* from its closest relatives *S. octosporus* and *S. cryophilus* around 120 million years ago (*Rhind et al., 2011*), and have not jumped across the species barriers. In contrast, *Spok* family genes have a patchy but widespread distribution among Pezizomycotina species (*Grognet et al., 2014*), whose last common ancestor dates to 400 million years ago (*Prieto and Wedin, 2013*), suggesting that *Spok* killers have originated much earlier than *wtf* killers or have been especially prone to transfer horizontally across species. Secondly, *spok* family genes encode soluble proteins with a C-terminal kinase-like domain (InterPro:IPR011009), whereas *wtf* genes are predicted to encode multi-transmembrane proteins, suggesting that the detailed molecular mechanisms of these two types of killers are likely to be quite distinct.

The single-gene nature of *wtf* killers begs the question: how does a single gene encode both the killing and the protecting activities? Our findings on the mutations that can disrupt one but not the other activity suggest the possibility that more than one type of expression products can be generated from a *wtf* killer gene. Consistent with this idea, Nuckolls et al. showed that the *wtf4* gene from the *S. pombe var. kambucha* strain (*Sk wtf4*) acts as a spore killer by expressing two overlapping transcripts, with the shorter transcript encoding a spore-killing poison and the longer transcript encoding an antidote that protects the spores from the killing effect of the poison (*Nuckolls et al., 2017*). This dual-transcript model can satisfactorily explain why the Tb and Te forms of *cw9* and *cw27* lost protecting activities but not killing activities (*Figure 4A*). Presumably, the transcription of the longer isoforms, which encode antidote proteins initiated from a start codon in the predicted exon 1, partially relies on the sequence between the upstream LTR and the conserved_up sequence and absolutely requires the conserved_up sequence; in contrast, the transcription of the shorter

isoforms, which encode poison proteins initiated from a downstream alternative start codon, is not affected by removing the upstream sequences.

We hypothesized that the transcription of the poison isoforms may depend on a special promoter residing within the predicted intron 1 of *cw9* and *cw27*. Indeed, through inspecting the DNA sequence alignment (*Figure 5—source data 1*), we found that *cw9*, *cw27*, and 10 other *wtf* genes in the reference genome and the CBS5557 genome share a 150 bp conserved sequence in the predicted intron 1, and this sequence is closely upstream of the transcription start site of a shorter-than-predicted *wtf23* transcript isoform uncovered by the Iso-Seq long-read RNA sequencing method (*Figure 5—figure supplement 3*) (*Kuang et al., 2017*). These 12 genes can be classified into two subtypes, which we call Intron-1-ATG genes (*wtf4*, *wtf13*, *cw4*, *cw18*, and *cw23*) and Exon-2-ATG genes (*wtf19*, *wtf23*, *cw8a*, *cw9*, *cw21*, *cw22*, and *cw27*), based on the locations of the first ATG codons downstream of the 150 bp sequence (*Figure 5C* and *Figure 5—figure supplement 3*). These ATG codons are all in frame with the predicted coding sequences. The within-intron-1 ATG codons in the 5 Intron-1-ATG genes are only 7 bp away from the predicted exon 2, and if used as start codons, would result in proteins beginning with three intron-1-coded amino acids (Met, Leu, and Ser). The two active spore killer genes found in the *S. pombe var. kambucha* strain by Nuckolls et al., *Sk wtf4* and *Sk wtf28*, also possess this 150 bp sequence and can be classified as an Intron-1-ATG gene and an Exon-2-ATG gene, respectively. We propose that this 150 bp sequence is a hallmark of the 'poison-and-antidote' genes that can, at least potentially, generate both poison and antidote isoforms according to the dual-transcript model proposed by Nuckolls et al.

The other 45 *wtf* genes in the reference genome and the CBS5557 genome lack this 150 bp sequence. In addition, except for the eight most divergent genes (*wtf7*, *cw7*, *wtf11*, *cw11*, *wtf14*, *cw14*, *wtf15*, and *cw15*), they do not have an in-frame ATG codon in intron one or in the 5'-section of exon 2, suggesting that most of these genes are not able to generate a shorter isoform according to the dual-transcript model, and are thus 'antidote-only' genes. Alternatively, it is possible that they may generate two different protein products using other mechanisms, such as alternative splicing (*Kuang et al., 2017*), overlapping proteins translated in different frames (*Duncan and Mata, 2014*), and post-translational processing and modification.

Five *wtf* genes in the reference genome, *wtf1*, *wtf2*, *wtf3*, *wtf22*, and *wtf24*, are annotated by PomBase as pseudogenes (*McDowall et al., 2015*). Our analysis of the gene structures of the 57 *wtf* genes in the reference genome and the CBS5557 genome showed that in addition to these five genes, four other *wtf* genes in the reference genome (*wtf6*, *wtf8*, *wtf12*, and *wtf17*), and three *wtf* genes in the CBS5557 genome (*cw3*, *cw12*, and *cw16*), appear to have suffered pseudogenizing mutations (see Materials and methods and *Figure 5—source data 2*), which include loss of start codons (*wtf3* and *cw3*), premature stop codons (*wtf1*, *wtf12*, *cw12*, and *cw16*), frameshifting indels (*wtf6*, *wtf8*, *wtf22*, and *wtf24*), and segmental deletions (*wtf2* and *wtf17*). These 'dead' *wtf* genes presumably are not able to generate either antidote or poison proteins.

Most of the *wtf* genes in the reference genome and the CBS5557 genome are predicted to contain either 5 or 6 exons. Interestingly, the 6-exon genes, including *cw9*, *cw27*, and all but one of the other ten 150-bp-sequence-containing genes, clustered together in the DNA sequence-based phylogenetic tree (*Figure 5C*), and their protein products are predicted to be longer and contain two or three more transmembrane helices than the protein products of 5-exon genes (*Figure 5—source data 3*, *Figure 5—figure supplement 4*, and *Figure 5—figure supplement 5*). For each of the 12 150-bp-sequence-containing genes, the antidote protein product is approximately 50 amino acids longer than the poison protein product at the N-terminal soluble region. The 10 C-terminal amino acids missing from the protein products of the Td forms of *cw9* and *cw27* are downstream of the last transmembrane helix, and are strongly conserved among the predicted protein products of 6-exon genes (*Figure 5—figure supplement 4*), indicating that Td-like mutations do not occur naturally to a significant extent.

How do transmembrane proteins act as killers? In the prokaryotic toxin-antitoxin (TA) systems, many toxins are transmembrane proteins and kill cells by damaging cellular membranes (*Schuster and Bertram, 2013*). The best-known examples include the type I TA toxins Hok and TisB, and the type V TA toxin GhoT (*Cheng et al., 2014*; *Gerdes et al., 1986*; *Unoson and Wagner, 2008*). We speculate that Wtf proteins may also possess abilities to disrupt the integrity of certain cellular membranes.

*wtf* genes were named after their association with the solo LTRs of the Tf retrotransposons (*Wood et al., 2002*). It was initially hypothesized that retrotransposition may aid the expansion of the *wtf* gene family (*Wood et al., 2002*). However, the presence of introns in *wtf* genes and the lack of sequence similarity between *wtf*-flanking LTRs indicate that *wtf* genes have not been retrotransposed (*Bowen et al., 2003*). Bowen et al. have put forward an explanation for the enrichment of LTRs near *wtf* genes: the regions flanking *wtf* genes may have been favored insertion targets of Tf retrotransposons, perhaps owing to the high transcription levels of *wtf* genes during meiosis and sporulation (*Bowen et al., 2003*). Our findings that LTRs flanking *cw9* and *cw27* are dispensable for spore killing support the idea that the *wtf*-LTR association does not benefit *wtf* genes but may rather reflect an exploitation of *wtf* genes by Tf retrotransposons for their own expression and/or inheritance benefit. Furthermore, we show that three singleton *wtf* genes (*cw26*, *cw27*, and *cw28*) appear to have been lost in the laboratory strain due to LTR-mediated recombination, suggesting that the *wtf*-LTR association promotes *wtf* gene loss and is detrimental to the long-term persistence of *wtf* genes. In addition, it may not be coincidental that both *cw9* and *cw27* require an approximately 90-bp-long sequence between the upstream LTR and the conserved_up sequence for the full protecting activity, raising the possibility that some *wtf* genes that lack such an intervening sequence (e.g. *wtf4* and *cw4*) may have suffered functional loss due to a Tf insertion too close to the conserved_up sequence or may have become functionally dependent on the upstream LTR.

In terms of the number of genes per genome, *wtf* genes are probably the most evolutionarily successful gamete killers known to date. During a relatively short evolutionary time span, *wtf* gene family has emerged and become the largest species-specific gene family in *S. pombe* (*Lespinet et al., 2002*). The compactness (minimal sizes of fully functional *cw9* and *cw27* are both around 1.9 kb) and the self-sufficiency of the *wtf* killers likely have been advantageous to its family size expansion. Furthermore, given the dramatic divergence of *wtf* genes between the reference genome and the CBS5557 genome, and the mutual-killing behaviors of *cw9* and *cw27*, it is possible that two identical copies of a *wtf* gene generated by a gene duplication event can undergo relatively rapid sequence diversification to become mutually non-resistant, so that their selective advantage is not diminished by resistance. Ectopic gene conversion is known to play a causal role in the sequence diversification of animal major histocompatibility complex (MHC) genes (*Ohta, 1991*; *Takuno et al., 2008*), plant disease-resistance genes (R genes) (*Mondragon-Palomino and Gaut, 2005*), and *Plasmodium falciparum var* genes (*Claessens et al., 2014*). Our phylogenetic analysis suggests that ectopic gene conversion may also contribute to the historical expansion and the ongoing dynamic change of the *wtf* gene family.

It is intriguing that most of the *wtf* genes locate on chromosome III, the smallest of the three fission yeast chromosomes. To explain this phenomenon, Bowen et al. proposed that the laboratory strain may have inherited only chromosome III but not the other two chromosomes from an ancestor strain in which *wtf* genes were abundant on all three chromosomes (*Bowen et al., 2003*). Our finding here of the same kind of chromosomal distribution bias in CBS5557, whose nucleotide diversity from the laboratory strain is higher on chromosome III than on the other two chromosomes (*Hu et al., 2015*), makes this explanation less likely. The only type of viable aneuploidy in fission yeast is chromosome III disomy (*Niwa et al., 2006*; *Niwa and Yanagida, 1985*), which occurs at a frequency of >1% during normal meiosis (*Molnar et al., 1995*). We speculate that the tolerance of chromosome III disomy may in some way be advantageous to the expansion of the *wtf* gene family on this chromosome.

Spore killers and other meiotic drivers, despite having only been found in a small number of taxa, are believed to exist in a wide range of species, where they are an important force influencing genome organization, gametogenesis, and speciation (*Burt and Trivers, 2006*; *Lindholm et al., 2016*; *Werren, 2011*). The molecular basis of their action and the evolutionary mechanisms behind their birth, spread, and extinction are scarcely known. The unveiling of the molecular identities of spore killers in *S. pombe* by this study and that of Nuckolls et al. (*Nuckolls et al., 2017*), promises to bring the power of a highly tractable model organism to the pursuit of the many mysteries of these evolutionary wonders.

## Materials and methods

### Strains

Fission yeast strains used in this study are listed in *Supplementary file 2* in Microsoft Word format. Genetic methods and composition of media are as described (*Forsburg and Rhind, 2006*). Gene deletion was constructed by PCR-based gene targeting. Integrating plasmids were based on pDUAL and related vectors (*Matsuyama et al., 2004*, *Matsuyama et al., 2008*). A pDUAL-based plasmid was linearized with NotI digestion and integrated at the *leu1* locus, or linearized with MluI digestion and integrated at the *ars1* replication origin region upstream of the *hus5* gene.

To convert homothallic strains to heterothallic and to mark the mating type locus, we created the *mat1Δ17* mutation (*Arcangioli and Klar, 1991*), by replacing a 140 bp sequence between the H1 homology box at the *mat1* locus and the nearby SspI restriction site with an antibiotic resistance marker using PCR-based gene targeting. $h^+/h^-$ diploid strains were constructed by mating on SPAS medium $h^+$ and $h^-$ strains with different antibiotic resistance markers inserted at the *mat1* locus, and a few hours later restreaking the mating mixture onto YES medium containing two types of antibiotics.

To construct strains with only a portion of chromosome III originating from CBS5557 and the rest of the genome coming from the laboratory strain, we first introduced *rec12Δ* mutation, which blocks meiotic recombination, into the laboratory strain and CBS5557, respectively. Through crossing the two resultant strains with each other, we obtained a strain whose chromosome I and II are from the laboratory strain and whose chromosome III is from CBS5557. This strain was then crossed to a *rec12⁺* laboratory-background strain carrying an antibiotic resistance marker at a locus on the left arm of chromosome III (*mug123:hphMX*) and another antibiotic resistance marker at a locus on the right arm of chromosome III (*rps2802:natMX*) (first backcross). Progeny strains with only one of the loci marked were kept. They were then crossed to laboratory-background strains with the other locus marked by *kanMX* (second backcross). By scoring the marker segregation ratios of the second backcross, we selected those first-backcross progenies that conferred strong segregation distortion at the *kanMX*-marked locus. Their second-backcross progenies that harbored the *kanMX* marker were tested by crossing, and those that conferred segregation distortion at the *kanMX*-marked locus were selected. One of the second-backcross progenies that conferred segregation distortion at the *mug123* locus is the backcrossed-1 strain used in the bulk segregant analysis shown in *Figure 1B*. Two chromosome III genomic regions from CBS5557, one between coordinates 247743 and 568805, and the other between coordinates 814348 and 1711635, are present in the backcrossed-1 strain. Another second-backcross progeny that conferred segregation distortion at the *rps2802* locus underwent additional rounds of backcross to yield the backcrossed-2 strain used in the bulk segregant analysis shown in *Figure 1C*. The CBS5557 chromosome III genomic region present in the backcrossed-2 strain is between coordinates 1839246 and 1916857.

### Bulk segregant analysis

For Illumina sequencing-assisted bulk segregant analysis, >150 viable progeny colonies derived from random spore analysis were pooled together. Genomic DNA extraction, sequencing libraries construction, and Illumina sequencing were performed as described (*Hu et al., 2015*). Sequencing data were deposited at NCBI SRA under the following accession numbers: viable progenies from CBS5557 × laboratory strain (DY9974 × DY8531), SRR5131575; viable progenies from backcrossed-1 × laboratory strain (cross-derived diploid strain was named DY26097), SRR5131578; viable progenies from backcrossed-1 *cw9Δ* × laboratory strain (cross-derived diploid strain was named DY26100), SRR5131579; viable progenies from backcrossed-2 strain × laboratory strain (cross-derived diploid strain was named DY26095), SRR5131577; viable progenies from backcrossed-2 *cw27Δ* strain × laboratory strain (cross-derived diploid strain was named DY26092), SRR5131576. Sequencing reads were mapped to the reference genome using BWA-MEM (version 0.7.15-r1140, RRID:SCR_010910). SAM files were converted to BAM files and duplicates were removed using SAMtools (version 0.1.19–44428 cd, RRID:SCR_002105). The average read depths were 14.9× for DY9974 × DY8531, 10.4× for DY26097, 18.4× for DY26100, 16.2× for DY26095, and 23.1× for DY26092. The software bam-readcount version 0.7.4 was run with the option -q 60 -b 35 (-q 60 -b 27 for DY26097, which had lower depth coverage and lower sequencing quality) to obtain the read

counts of four different bases at each of the 38783 SNP positions identified previously (*Hu et al., 2015*). Reference allele frequency at each SNP position was calculated by dividing the read count of the reference base by the sum of the read counts of the reference base and the variant base. Only SNP positions with the sum of the read counts of the reference base and the variant base >= 10 (> = 7 for DY26097) were used in the plots. A trend line was drawn in each plot by calculating a rolling median of reference allele frequencies at 45 consecutive SNP positions using the rollapply function of R's zoo package (version 1.7–13).

## PacBio sequencing

For PacBio sequencing of CBS5557 genome, we prepared genomic DNA from DY9971, an $h^-$ derivative of CBS5557, by first grinding cells in liquid nitrogen and then extracting DNA using the Maxi Column Fungal DNAOUT kit (Tiandz, Beijing, China). Sequencing library construction and sequencing on the PacBio RS II platform using P6C4 chemistry were performed by BGI (Shenzhen, China). After filtering out low-quality reads and removing adaptor sequences, we obtained 91,333 subreads no shorter than 1000 bp (mean length 7370 bp). These reads were used for de novo assembly by the SMRT Analysis software. The assembly was polished using Pilon version 1.21 (RRID:SCR_014731) together with our previously published CBS5557 Illumina sequencing data (*Hu et al., 2015*; *Walker et al., 2014*). PacBio sequencing data were deposited at NCBI SRA under the accession number SRR5133273.

## Electron microscopy analysis

For the induction of synchronous meiosis and sporulation, $h^+/h^-$ diploid strains were pre-grown in liquid YES medium to log phase, cultured in SSL+N synthetic liquid medium for 10 hr, and then shifted to nitrogen-free SSL-N medium (*Egel and Egel-Mitani, 1974*). Four-spored asci represented about half of the cells 10 hr after shifting to SSL-N (*Olson et al., 1978*). Electron microscopy was performed as described (*Sun et al., 2013*).

## Spore viability analysis

Spore viability was assessed by tetrad analysis using a TDM50 tetrad dissection microscope (Micro Video Instruments, Avon, USA). At least 40 tetrads (160 spores) were analyzed for each cross. Based on empirical evidence, such a sample size is large enough for reliably assessing spore viability. Numerical data of the tetrad analysis in Excel format are provided as *Supplementary file 1*. For statistical analysis of the spore viability data, Fisher's exact test and exact binomial test of goodness-of-fit were performed using Excel spreadsheets downloaded from http://www.biostathandbook.com/fishers.html and http://www.biostathandbook.com/exactgof.html, respectively (*McDonald, 2014*).

## Sequence and phylogenetic analysis

Except for the phylogenetic analysis, DNA sequences were aligned using MAFFT via Jalview (*Katoh et al., 2002*; *Waterhouse et al., 2009*). Pair-wise identity was calculated by dividing the numbers of identical bases by the alignment length using the Sequence Manipulation Suite web server (http://www.bioinformatics.org/sms2/ident_sim.html) (*Stothard, 2000*).

For the phylogenetic analysis shown in *Figure 5C*, an alignment of DNA sequences containing the conserved_up regions, the coding sequences, and the associated introns was generated using the L-INS-i algorithm of MAFFT version 7.310 for Mac OS X (RRID:SCR_011811) with the command line option –localpair –maxiterate 16 –reorder (*Katoh and Standley, 2014*), and manually adjusted to correct an obvious misalignment of the *wtf2* sequence. The final alignment of 57 *wtf* genes of the reference and CBS5557 genome in FASTA format is provided as *Figure 5—source data 1*. Maximum likelihood analysis was performed using IQ-TREE version 1.5.3 for Mac OS X with the command line option -m TEST -alrt 1000 -bb 1000 (*Nguyen et al., 2015*). The tree was rooted by midpoint rooting using FigTree version 1.4.2 (http://tree.bio.ed.ac.uk/software/figtree/) and was visualized using Phylo.io (*Robinson et al., 2016*).

Gene structures of the 57 *wtf* genes were predicted using the AUGUSTUS web server (version 3.2.3, RRID:SCR_008417) with the parameter settings –genemodel=exactlyone –sample=100 –keep_viterbi=true –alternatives-from-sampling=true –minexonintronprob=0.08 –minmeanexonintronprob=0.3 –maxtracks=20 (*Stanke et al., 2008*). When multiple predications were made for a

gene, we manually inspected the predictions and selected the one that best conforms to conserved patterns of intron positions. For the majority of genes, the selected prediction is the top-scoring prediction, and for 10 genes (*wtf6*, *wtf7*, *wtf14*, *wtf24*, *cw6a*, *cw7*, *cw12*, *cw14*, *cw16*, *cw25*), the selected prediction is the one with the second best score. The final predictions for all 57 genes in plain text format are provided as *Figure 5—source data 2*, which can be loaded into Jalview as an annotation file for the DNA sequence alignment in *Figure 5—source data 1*. The failure to predict an exon flanked by conserved intron-exon junction sequences is an indication that pseudogenizing mutation(s) exist. Among the predicted protein sequences obtained based on these gene structure predictions, the sequences of Wtf4, Wtf6, Wtf7, Wtf12, Wtf13, Wtf15, Wtf17, and Wtf18 are different from those annotated at PomBase. A MAFFT (L-INS-i algorithm)-generated alignment of the 57 predicted protein sequences in FASTA format is provided as *Figure 5—source data 3*.

## Data availability

Illumina and PacBio sequencing data reported in this paper are available at the Sequence Read Archive (SRA) under the accession number SRP095878 (BioProject PRJNA358837) (*Hu et al., 2017*). The DNA sequences and annotations of the 32 *wtf* genes of the CBS5557 genome have been deposited at GenBank under the accession numbers KY926712-KY926743.

## Acknowledgements

We thank Sarah Zanders for communicating results prior to publication. We are grateful to Meng-Qiu Dong for critical reading of the manuscript. This work was supported by the National Basic Research Program of China (973 Program, 2014CB849901).

## Additional information

### Competing interests

WH, FS, L-LD: A patent application related to this work has been filed. The other authors declare that no competing interests exist.

### Funding

| Funder | Grant reference number | Author |
| --- | --- | --- |
| Ministry of Science and Technology of the People's Republic of China | National Basic Research Program of China 2014CB849901 | Li-Lin Du |

The funders had no role in study design, data collection and interpretation, or the decision to submit the work for publication.

### Author contributions

WH, Conceptualization, Investigation, Writing—original draft; Z-DJ, J-XZ, Investigation; FS, Formal analysis, Writing—original draft; W-ZH, Formal analysis; L-LD, Conceptualization, Funding acquisition, Writing—original draft, Writing—review and editing

### Author ORCIDs

Wen Hu, http://orcid.org/0000-0002-0604-2119
Li-Lin Du, http://orcid.org/0000-0002-1028-7397

## Additional files

### Supplementary files

• Supplementary file 1 Numerical data of the tetrad analysis.

• Supplementary file 2 Fission yeast strains used in this study.

## Major datasets

The following dataset was generated:

| Author(s) | Year | Dataset title | Dataset URL | Database, license, and accessibility information |
|---|---|---|---|---|
| Hu W, Suo F, Du LL | 2017 | Schizosaccharomyces pombe Genome sequencing | https://www.ncbi.nlm.nih.gov/bioproject/PRJNA358837 | Publicly available at the NCBI BioProject (accession no: PRJNA358837) |

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
