## [Decision Letter]

Thank you for submitting your article "A large gene family in fission yeast encodes spore killers that subvert Mendel's law" for consideration by *eLife*. Your article has been favorably evaluated by Diethard Tautz (Senior Editor) and three reviewers, one of whom is a member of our Board of Reviewing Editors. The following individuals involved in review of your submission have agreed to reveal their identity: Nick Rhind (Reviewer #2); Michael Lichten (Reviewer #3).

The reviewers have discussed the reviews with one another and the Reviewing Editor has drafted this decision to help you prepare a revised submission.

Summary:

Hu et al. have identified, in the genome of *S. pombe* strain CBS5557 two members of the *wtf* repeated gene family that serve as meiotic drive elements when crossed to the reference strain. Using deletions of the two loci, they show that the protective function can be separated from the killing function, and show that the C-terminus of the putative open-reading frame is important for killing but not for protection. In addition, they show that the two genes actually kill their sister spores, using both viability and EM-based assays. They then analyze the ensemble of *wtf/cw* loci in CBS5557 and show that these elements appear to frequently "move" by ectopic gene conversion. Combined with Nuckolls et al., this paper makes an intriguing and interesting story.

Essential revisions:

1) The switch in nomenclature, from *k1wtf* and *k2wtf* to *cw9* and *cw27*, is needlessly confusing. Why not simply call these loci *cw9* and *cw27* from the start, and just say that the reasons for doing so will be presented below? Along similar lines, the *wtf* nomenclature depicted in Figure 5 seems unnecessarily complicated. In situations in which similarity and synteny make homology unambiguous, ambiguous nomenclature is not useful. For instance, *cw[3-4]a* and *cw[3-4]b* should be named *cw3* and *cw4*, respectively, likewise the homologs of *wtf11, 12, 14, 15, 17* and *18* are unambiguous.

2) Figure 1 is confusing, especially since it hides the fact that there actually are no SNPs segregating in the regions where the red line is at 1. Why not simply display the rolling median for regions where there actually are SNPs? The first and last 40 SNPs will not be displayed, but there appear to be so many that this will not materially affect the presentation. The terms "left-peak" and "right-peak" strain are also confusing; we suggest adding a panel that shows a chr 3 map for the two strains with the reference and CBS5557-introgressed sequences in different colors, and including that color code on the X-axis of the plots in panel B.

3) It's not clear why *k1wft*∆ does not restore reference allele recovery to 50% in the left-peak strain x laboratory strain cross (Figure 1). Is it possible that *cw[11-12]c* also has drive activity?

4) Figure 5, panel C would be considerably more useful if it contained a horizontal scale, to roughly indicate the extent of divergence between different alleles. Criteria for considering two alleles to be paired should also be indicated; for example, it's not clear why *cw8a* and *cw24* are considered to be a homeologous pair, but *wtf24* and *cw6b* are not. Furthermore, the phylogeny should be shown as an unrooted tree – the authors reconstructed the phylogeny as unrooted and they should depict it that way too.

5) For readers interested in further numerical analysis, it would be useful if the actual numerical data underlying all of the bar graphs were included in a supplementary file. For example, we were wondering if the loss of spore viability in *k1*∆ *k2*∆ double mutant spores (Figure 2) is greater than what one would expect on an additive basis of spore viability loss in the two single mutant spore types-it looks as if it is, but with the current presentation it's impossible to check.

6) It would be useful to use the translation start site-criterion outlined in Nuckolls et al. for whether the *cw* loci are "dead", "antidote-only", or "antidote-toxin". Similarly, since the C-termini of *cw9* and *cw27* appear to be important for toxin activity, is this region more conserved among the wtf loci than other parts of the protein? In "antidote-only" or in "antidote-toxin" loci?

7) It is not clear why the variation of allele frequency depicted in Figure 1 is so large. If over 150 segregants were sequenced, how can some loci in regions that segregate 50:50 (e.g. on the left end of Chromosome I) be represented by only one allele?

8) The authors should extend the discussion of their findings and take into account the findings of Nuckolls et al. In particular:

What do the other *wtf* genes do? Could they be acting as rescuers but not as killers? Could they have assumed other unrelated functions?

Figure 5 shows an unusual concentration of *wtf* genes in chr III. Why is that the case? (Could it have to do with the fact that it is the only chromosome where disomy is tolerated?)

On the second page of the Discussion, the authors speculate as to how one locus could produce both a toxin and an antidote. It should be possible to eliminate, or at least deprioritize, one or more of the proposed mechanisms by inspecting the sequence of the loci and perhaps publicly available RNA-seq data. It would be of great interest to have some speculation on how a putative trans-membrane protein could function in a toxin-antidote system.

---

## [Author Response]

Essential revisions:

1) The switch in nomenclature, from k1wtf and k2wtf to cw9 and cw27, is needlessly confusing. Why not simply call these loci cw9 and cw27 from the start, and just say that the reasons for doing so will be presented below? Along similar lines, the wtf nomenclature depicted in Figure 5 seems unnecessarily complicated. In situations in which similarity and synteny make homology unambiguous, ambiguous nomenclature is not useful. For instance, cw[3-4]a and cw[3-4]b should be named cw3 and cw4, respectively, likewise the homologs of wtf11, 12, 14, 15, 17 and 18 are unambiguous.

We have followed the suggestions of the reviewers and changed the nomenclature of the *wtf* genes in the CBS5557 genome.

2) Figure 1 is confusing, especially since it hides the fact that there actually are no SNPs segregating in the regions where the red line is at 1. Why not simply display the rolling median for regions where there actually are SNPs? The first and last 40 SNPs will not be displayed, but there appear to be so many that this will not materially affect the presentation. The terms "left-peak" and "right-peak" strain are also confusing; we suggest adding a panel that shows a chr 3 map for the two strains with the reference and CBS5557-introgressed sequences in different colors, and including that color code on the X-axis of the plots in panel B.

Figure 1 have been revised according to the suggestions of the reviewers. We have renamed the two backcrossed strains backcrossed-1 and backcrossed-2, respectively.

3) It's not clear why k1wft∆ does not restore reference allele recovery to 50% in the left-peak strain x laboratory strain cross (Figure 1). Is it possible that cw[11-12]c also has drive activity?

We agree with the reviewers that the residual allele frequency bias is probably due to additional active spore killer gene(s) in the CBS5557 genomic regions still present in the backcrossed-1 strain. We have added a sentence about this possible explanation in the Results section. There are 12 *cw* genes remaining in the backcrossed-1 strain. Future work will be needed to determine which one(s) are active killer genes.

4) Figure 5, panel C would be considerably more useful if it contained a horizontal scale, to roughly indicate the extent of divergence between different alleles. Criteria for considering two alleles to be paired should also be indicated; for example, it's not clear why cw8a and cw24 are considered to be a homeologous pair, but wtf24 and cw6b are not. Furthermore, the phylogeny should be shown as an unrooted tree – the authors reconstructed the phylogeny as unrooted and they should depict it that way too.

During the revision, we have improved the de novo assembly of CBS5557 genome by adding a polishing step using the software Pilon. With polished sequences of the *cw* genes, we have generated a new phylogenetic tree, which has a topology similar to, but not exactly the same as, the tree in the original manuscript. This new tree is shown in Figure 5 of the revised manuscript. We now include a scale bar in Figure 5, as suggested by the reviewers. In the Results section and in the legend of Figure 5, we have now more precisely defined the criteria for highlighting a pair of genes, which we now term phylogenetic neighbors. Two genes are considered phylogenetic neighbors if they are separated on the phylogenetic tree by a single internal node with an IQ-tree-calculated ultrafast bootstrap support value >= 95%. We agree with the reviewers that it is inappropriate to draw unrooted tree in a rooted format. We have now rooted the tree using midpoint rooting, which is a widely used method for rooting a phylogenetic tree when no suitable outgroup is available (see the reference Hess & De Moraes Russo, 2007).

5) For readers interested in further numerical analysis, it would be useful if the actual numerical data underlying all of the bar graphs were included in a supplementary file. For example, we were wondering if the loss of spore viability in k1∆ k2∆ double mutant spores (Figure 2) is greater than what one would expect on an additive basis of spore viability loss in the two single mutant spore types-it looks as if it is, but with the current presentation it's impossible to check.

Numerical data are now provided in [Supplementary-material SD4-data] of the revised manuscript. The double mutant spores indeed appeared to have suffered a greater loss of viability than expected from a simple additive effect, but the exact reason is not clear at this moment.

6) It would be useful to use the translation start site-criterion outlined in Nuckolls et al. for whether the cw loci are "dead", "antidote-only", or "antidote-toxin". Similarly, since the C-termini of cw9 and cw27 appear to be important for toxin activity, is this region more conserved among the wtf loci than other parts of the protein? In "antidote-only" or in "antidote-toxin" loci?

We have more carefully examined the sequences of the 57 *wtf* genes in the reference genome and the CBS5557 genome, and uncovered a within-intron-1 150-bp sequence shared by all currently known active *wtf* killers (*cw9* and *cw27* found in our study and *Sk wtf4* and *Sk wtf28* found by Nuckolls et al.). We added a new figure (Figure 5—figure supplement 3) about this sequence. We propose that this sequence may serve as a promoter driving the transcription of the poison isoforms and the presence of this sequence is a useful criterion for identifying "poison-and-antidote" genes. In the Discussion section of the revised manuscript, we added three paragraphs to discuss "poison-and-antidote" genes that share this 150-bp sequence, "antidote-only" genes lacking this 150-bp sequence, and the "dead" genes that have suffered pseudogenizing mutations, respectively.

Based on the predicted gene structures of the *wtf* genes, most of them have either 5 or 6 exons. All currently known active *wtf* killers are 6-exon genes. We have added two new figures (Figure 5—figure supplement 4 and Figure 5—figure supplement 5) depicting the sequence alignments of the protein products of the 6-exon genes and the 5-exon genes, respectively. From the alignment, it can be seen that the C-terminal sequences of Cw9 and Cw27 proteins are strongly conserved among the protein products of 6-exon genes, regardless of whether they are "poison-and-antidote" genes or "antidote-only" genes.

7) It is not clear why the variation of allele frequency depicted in Figure 1 is so large. If over 150 segregants were sequenced, how can some loci in regions that segregate 50:50 (e.g. on the left end of Chromosome I) be represented by only one allele?

For cost reasons, we usually limit the Illumina sequencing of the bulk segregant analysis (BSA) samples to an average read depth of between 10× and 20×. Therefore, the read depth at any SNP loci is much lower than the number of segregants, and as a consequence, the allele frequency data shown in Figure 1 are much noisier than expected from the number of segregants. We have now listed in the Materials and methods section the average read depths of the five BSA sequencing datasets.

*8) The authors should extend the discussion of their findings and take into account the findings of Nuckolls et al. In particular:*

What do the other wtf genes do? Could they be acting as rescuers but not as killers? Could they have assumed other unrelated functions?

We have added several paragraphs in the Discussion section to describe the classification the 57 *wtf* genes in the reference genome and the CBS5557 genome into "poison-and-antidote" genes, "antidote-only" genes, and "dead" genes.

Figure 5 shows an unusual concentration of wtf genes in chr III. Why is that the case? (Could it have to do with the fact that it is the only chromosome where disomy is tolerated?)

We agree with the reviewers that tolerance of chromosome III disomy may be a reason why *wtf* genes concentrate on chromosome III. We have added a paragraph in the Discussion section to discuss the possible reasons behind this intriguing phenomenon.

On the second page of the Discussion, the authors speculate as to how one locus could produce both a toxin and an antidote. It should be possible to eliminate, or at least deprioritize, one or more of the proposed mechanisms by inspecting the sequence of the loci and perhaps publicly available RNA-seq data. It would be of great interest to have some speculation on how a putative trans-membrane protein could function in a toxin-antidote system.

In the newly added paragraphs of the Discussion section, by more carefully inspecting the sequences, we have classified the 57 *wtf* genes in the reference genome and the CBS5557 genome into "poison-and-antidote" genes, which conform to the dual-transcript model of Nuckolls et al., "antidote-only" genes, and "dead" genes. We speculate that it remains possible that the "antidote-only" genes may be able to generate poison isoforms through mechanisms other than alternative transcription start sites. We have also added a paragraph in the Discussion section to speculate on how transmembrane proteins can act as spore killers.